# A Sub-Problem Quantum Alternating Operator Ansatz for Correlation Clustering

**Lucas Fabian Naumann** [1]   **Jannik Irmai** [1]   **Bjoern Andres** [1,2]

## Abstract

The Quantum Alternating Operator Ansatz (QAOA) is a hybrid quantum-classical variational algorithm for approximately solving combinatorial optimization problems on Noisy Intermediate-Scale Quantum (NISQ) devices. Although it has been successfully applied to a variety of problems, there is only limited work on correlation clustering due to the difficulty of modelling the problem constraints with the ansatz. Motivated by this, we present a generalization of QAOA that is more suitable for this problem. In particular, we modify QAOA in two ways: Firstly, we use nucleus sampling for the computation of the expected cost. Secondly, we split the problem into sub-problems, solving each individually with QAOA. We call this generalization the Sub-Problem Quantum Alternating Operator Ansatz (SQAOA) and show theoretically that optimal solutions to correlation clustering instances can be obtained with certainty when the depth of the ansatz tends to infinity. Further, we show experimentally that SQAOA achieves better approximation ratios than QAOA for correlation clustering, while using only one qubit per node of the respective problem instance and reducing the runtime (of simulations).

## 1. Introduction

The term "quantum supremacy" (Preskill, 2012) refers to the ability of quantum computers to perform tasks efficiently that classical computers cannot. From a theoretical point of view, algorithms achieving this supremacy for problems of practical interest have long been established. However, applying these algorithms to problem instances of classically intractable sizes is not possible on current quantum computers, as these are limited both by their number of qubits and the number of operations that can be performed on a qubit before its state is too corrupted by noise (circuit depth). These current quantum computers are also referred to as Noisy Intermediate-Scale Quantum (NISQ) devices (Preskill, 2018).

Variational quantum algorithms (Cerezo et al., 2021) have emerged as a promising paradigm to achieve quantum supremacy for practical problems on NISQ devices. These algorithms combine quantum and classical computing by using parameterized quantum circuits with few qubits and low depth, whose parameters are learned in a classical optimization loop. The Quantum Alternating Operator Ansatz (QAOA) (Hadfield et al., 2019) is such a variational algorithm designed for approximately solving combinatorial optimization problems. In particular, it alternately applies a parameterized phase-separation operator, which changes the phase of states depending on their cost, and a mixing operator, which enables transitions between states and thus constructive or destructive interference based on their phase difference. The number of times $p$ these operators are applied alternately is called the ansatz depth. For $p \to \infty$, and under the conditions given by Binkowski et al. (2024), there exist parameters for each problem instance such that QAOA returns an optimal solution with certainty. QAOA has been applied to a variety of problems (Cook et al., 2020; Saleem, 2020; Tabi et al., 2020; Fuchs et al., 2021), including correlation clustering (Weggemans, 2020; Weggemans et al., 2022).

Correlation clustering (Bansal et al., 2004) is a special clustering formulation in which objects are represented by the nodes of a graph, (dis-)similarities between them by edges with corresponding costs, and the goal is to cluster the nodes of the graph such that a cost function is optimized. In prominent difference to other formulations, the number of clusters is not fixed in advance, but learned from the data. This unsupervised clustering of objects based solely on pairwise (dis-)similarities finds application in various domains, such as computational biology (D'haeseleer, 2005; Erola et al., 2020), data analysis (Benjelloun et al., 2009; Abbas & Swoboda, 2023) and image segmentation (Yarkony et al., 2012; Beier et al., 2015; Keuper et al., 2015).

---

[1]Faculty of Computer Science, TU Dresden [2]Center for Scalable Data Analytics and AI, Dresden/Leipzig. Correspondence to: Bjoern Andres <bjoern.andres@tu-dresden.de>.

*Proceedings of the $42^{nd}$ International Conference on Machine Learning*, Vancouver, Canada. PMLR 267, 2025. Copyright 2025 by the author(s).

There are different variants of correlation clustering with respect to the associated costs and the cost function. In this article, we consider unweighted ($\{+1, -1\}$ costs) maximum agreement correlation clustering and will assume that, unless otherwise specified, the term "correlation clustering" refers to this variant. However, our approach can easily be adapted to weighted correlation clustering and other cost functions.

We introduce the Sub-Problem Quantum Alternating Operator Ansatz (SQAOA), a generalization of QAOA motivated by the application to correlation clustering and based on the idea of nucleus sampling (Holtzman et al., 2020) and the concept of splitting a problem into several dependent sub-problems. For a specific SQAOA formulation of correlation clustering we show that:

- For each instance, there exist parameters such that an optimal solution is obtained with certainty for $p \to \infty$.

- Only as many qubits are required to solve an instance as there are elements to cluster.

- It experimentally outperforms existing approaches in terms of approximation ratios and runtimes on complete and Erdős-Rényi graphs with up to 10 nodes.

## 2. Related Work

Unweighted maximum agreement correlation clustering on general graphs is known to be APX-hard. In particular, it is NP-hard for every $\epsilon > 0$ to approximate the problem within a factor of $80/79 - \epsilon$ (Tan, 2008). The best known classical algorithm for approximating unweighted maximum agreement correlation clustering is given by Swamy (2004) and achieves an approximation ratio of 0.7666 (Swamy bound). However, there exists a polynomial time approximation scheme when restricting the considered graphs to be complete (Bansal et al., 2004).

The only other work applying QAOA to a correlation clustering variant is by Weggemans (2020) and Weggemans et al. (2022). Weggemans (2020) reviews different QAOA formulations for correlation clustering with respect to the number of used qubits, circuit complexities and approximation ratios obtained by simulations. Furthermore, improvement strategies for the standard QAOA algorithm are evaluated, like the choice of the classical optimizer, the choice of initial parameters and the number of restarts. Most importantly, it is found that the achieved approximation ratios can be significantly increased by looping over the cluster number, i.e., by applying QAOA repeatedly, varying the number of allowed clusters from 1 to the number of nodes in the graph and returning only the best result.

From these studies, a "multi-level" formulation emerges as the best approach, in which each element to be clustered is associated with a qudit and the cluster of that element is given by the state of the qudit. We will use this formulation as a reference to benchmark SQAOA against. Using techniques from Farhi et al. (2014) and Wurtz & Love (2021), it is further shown for this multi-level QAOA formulation (including looping over the cluster number) that, for $p = 1$, there exist parameters achieving an approximation ratio of at least 0.6367 on all 3-regular graphs. Weggemans et al. (2022) build on this work by extending the evaluation of the multi-level formulation and describing how to realize it on a Rydberg system. However, the described implementation is restricted to 4-level qudits, i.e., qudits with four states, such that only solutions involving at most 4 clusters can be considered.

There is a variety of generalizations and variations of QAOA; a recent survey is by Blekos et al. (2024). However, to our knowledge, there exists no work on using different sampling strategies to compute the expected cost. And although there are approaches that apply QAOA to sub-problems (Tomesh et al., 2022; Esposito & Danzig, 2024), these split a problem instance into smaller instances of the same problem that are solved independently. In contrast, we solve instances of dependent sub-problems that are different from the original problem.

## 3. Preliminaries

We begin this section with a brief review of the notation and the fundamentals of quantum computing to the extent necessary for understanding the article. A thorough introduction can be found, e.g., in Nielsen & Chuang (2010). We then formally describe the Quantum Alternating Operator Ansatz and the correlation clustering problem before introducing our SQAOA formulation for correlation clustering in the next section.

**Notation and Fundamentals of Quantum Computing** We use the Dirac notation, i.e., we denote elements of $\mathbb{C}^n$ by $|\cdot\rangle$, their conjugate transpose by $\langle\cdot|$ and write $\langle x|y\rangle := \langle x| \, |y\rangle = |x\rangle^\dagger \, |y\rangle$ for the inner product of $|x\rangle, |y\rangle \in \mathbb{C}^n$.

We denote the standard unit vectors of $\mathbb{C}^2$ by

$$|0\rangle = \begin{bmatrix} 1 \\ 0 \end{bmatrix} \quad \text{and} \quad |1\rangle = \begin{bmatrix} 0 \\ 1 \end{bmatrix}.$$

Consequently, the standard unit vectors of $\mathbb{C}^{2^n} = \bigotimes_{j=1}^n \mathbb{C}^2$ are given by

$$\bigotimes_{j=1}^n |x\rangle \quad \text{where} \quad |x\rangle \in \{|0\rangle, |1\rangle\},$$

for which we introduce the abbreviation

$$|x\rangle \quad \text{where} \quad x \in \{0, 1\}^n.$$

The state of an $n$-qubit system is given by a normalized vector (statevector) in the Hilbert space $\mathbb{C}^{2^n} = \bigotimes_{j=1}^{n} \mathbb{C}^2$, which can be written as

$$|\psi\rangle = \sum_{x \in \{0,1\}^n} a_x |x\rangle \quad \text{with} \quad ||\psi\rangle|^2 = \sum_{x \in \{0,1\}^n} |a_x|^2 = 1 \,.$$

When measuring such a system (with respect to the computational basis), it collapses to one of the computational basis states $|x\rangle$. The coefficients $a_x$ are called probability amplitudes and their squared absolutes $|a_x|^2$ give the probability of collapsing into state $|x\rangle$.

In gate-based quantum computing, algorithms are realized by manipulating qubits with quantum gates. Quantum gates acting on $n$ qubits can be represented by unitary matrices of size $2^n$. Another way of characterizing quantum gates is by Hermitian matrices. For any unitary matrix $U$, there exists a Hermitian matrix $H$ such that $U = e^{-iH}$, and vice versa. In the remainder of this article, we will assume that matrices denoted by $U$ are unitary and that matrices denoted by $H$ are Hermitian. An overview of the gates used in this article is given in Appendix A. We use the following notation for applying a unitary $U$ operating on a single qubit to the $j$-th qubit of an $n$-qubit system:

$$U_j := \left( \bigotimes_{k=1}^{j-1} I \right) \otimes U \otimes \left( \bigotimes_{k=j+1}^{n} I \right) .$$

Quantum states differ from probability distributions as their probability amplitudes can take complex values. This means that, in addition to their absolute value, quantum states have a phase. This fact enables constructive and destructive interference between them, i.e., applying an operation to a quantum state is different from just applying it to all of its basis states separately.

**Quantum Alternating Operator Ansatz** The Quantum Alternating Operator Ansatz (QAOA) (Hadfield et al., 2019) is a variational algorithm for approximately solving combinatorial optimization problems. It generalizes the Quantum Approximate Optimization Algorithm (Farhi et al., 2014) which is, on the other hand, a translation of the Quantum Adiabatic Algorithm (Farhi et al., 2001) from adiabatic quantum computing to gate-based quantum computing.

A generic combinatorial optimization problem of size $n$ with feasible space $S \subseteq \{0,1\}^n$ and cost function $C : S \rightarrow \mathbb{R}$ can be written as

$$\min_{x \in S} C(x) \,.$$

Applying QAOA of depth $p$ for solving this problem, we first compute

$$|\boldsymbol{\beta}, \boldsymbol{\gamma}\rangle^{\text{QAOA}} := \left( \prod_{j=1}^{p} U_M(\beta_j) U_C(\gamma_j) \right) |s\rangle \,, \quad (1)$$

where

- $|s\rangle$ is an initial state in the feasible space $\mathcal{S}$, which is given by the set of all superpositions of classically feasible states, i.e., by

$$\mathcal{S} := \left\{ \sum_i \lambda_i |x\rangle \,\middle|\, \sum_i |\lambda_i|^2 = 1, \lambda_i \in \mathbb{C}, x \in S \right\} ;$$

- $U_C(\gamma) = e^{-i\gamma H_C}$ is a phase-separation operator such that

$$H_C |x\rangle = C(x) |x\rangle \,, \quad (2)$$

and $H_C$ is called cost Hamiltonian;

- $U_M(\beta) = e^{-i\beta H_M}$ is a mixing operator that preserves feasible states

$$\forall |\psi\rangle \in \mathcal{S} \, \forall \beta \in \mathbb{R} : \quad U_M(\beta) |\psi\rangle \in \mathcal{S} \,, \quad (3)$$

and allows for full mixing of solutions

$$\forall x, y \in S \, \exists \beta \in \mathbb{R} \, \exists r \in \mathbb{N} :$$
$$\langle y| \, U_M^r(\beta) \, |x\rangle > 0 \; ; \quad (4)$$

The parameters $\boldsymbol{\beta}, \boldsymbol{\gamma}$ are learned in a classical optimization loop such that the expectation value of the cost function is minimized

$$\langle \boldsymbol{\beta}, \boldsymbol{\gamma}|^{\text{QAOA}} \, H_C \, |\boldsymbol{\beta}, \boldsymbol{\gamma}\rangle^{\text{QAOA}} \,. \quad (5)$$

The intuition behind this ansatz is that the phase-separation operator modifies the phase of basis states (which correspond to feasible solutions) depending on their cost, while the mixing operator realizes transitions between feasible states, resulting in constructive and destructive interference. Since the parameters of the operators are optimized with respect to the expectation value of the cost function, states with low cost are amplified by constructive interference, while states with high cost are erased by destructive interference.

In order to apply QAOA to a specific problem, the operators and the initial state need to be defined and implemented. The main challenge lies in constructing the initial state and the mixing operator. Conversely, the phase-separation operator is easy to construct. If the problem is formulated as an integer program with binary variables, it is sufficient to choose the cost Hamiltonian $H_C$ such that variables $x_i$ in the cost function $C(x)$ are replaced by the term $\frac{(1-Z_i)}{2}$. This is due to the fact that if $x_i = 0$, then $\frac{(1-Z_i)}{2} |x_i\rangle = 0 |x_i\rangle$, and if $x_i = 1$, then $\frac{(1-Z_i)}{2} |x_i\rangle = 1 |x_i\rangle$. Thus, (2) is fulfilled for the Hamiltonian constructed in this way. Implementing the corresponding unitary operator only requires the application of RZ gates to individual qubits.

QAOA is considered a promising variational quantum algorithm for the following reasons:

- For $p \rightarrow \infty$ and under the conditions given in (Binkowski et al., 2024), there exist parameters for each problem instance such that an optimal solution is obtained with certainty.

- Under reasonable complexity theoretic conjectures, it is not possible to efficiently sample from the generated distributions classically, even for $p = 1$ (Farhi & Harrow, 2019).

- The parameters are concentrated for different instances of the same problem (Brandao et al., 2018; Akshay et al., 2021), allowing us to learn them for one instance and reuse them, or use them as an initial point for others.

The approximation ratios are expected to increase with larger ansatz depth $p$ and are guaranteed to improve with optimal parameters. However, the depth is limited for two reasons. Firstly, the number of applied operators increases, resulting in problems with computational resources for the simulation on classical computers and the introduction of noise for the execution on quantum computers. Secondly, the number of learnable parameters increases with $p$, and gradients cannot be easily computed on quantum computers.

**Correlation Clustering** Let $G = (V, E)$ be an undirected graph, let $n = |V|$ be the number of nodes in $G$ and let $c \in \{+1, -1\}^E$ be costs associated with the edges of the graph. The problem of unweighted maximum agreement correlation clustering consists in finding a clustering (or partition) of the node set $V$ such that the number of pairs of nodes connected by edges with cost $+1$ that are in the same cluster, and the number of pairs of nodes connected by edges with cost $-1$ that are in different clusters, is maximized. Figure 1 shows an example of a problem instance.

We can formulate unweighted maximum agreement correlation clustering as an integer quadratic program in which binary variables $x \in \{0, 1\}^{n \times n}$ indicate if a node $v$ is assigned to cluster $i$, $x_{v,i} = 1$, or not, $x_{v,i} = 0$:

$$
\begin{aligned}
\max_{x} \quad & \sum_{uv \in E \,:\, c_{uv}=1} \sum_{i \in K} x_{u,i}\, x_{v,i} + \\
& \sum_{uv \in E \,:\, c_{uv}=-1} \sum_{i,j \in K \,:\, i \neq j} x_{u,i}\, x_{v,j} \quad (6) \\
\text{subject to} \quad & \sum_{i \in K} x_{u,i} = 1 \quad \text{for all } u \in V,
\end{aligned}
$$

where $K = \{1, \ldots, n\}$ and we use $uv$, $vu$ for denoting an edge $\{u, v\} \in E$.

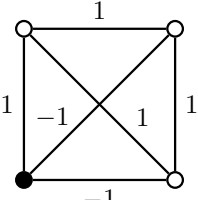

Figure 1. Depicted above is an example of an instance of the unweighted maximum agreement correlation clustering problem, along with a corresponding optimal solution. The problem instance is given by the graph and the costs written along its edges. The clustering indicated by the coloring of the nodes has 5 agreements and is optimal.

In this formulation, the variable assignment $x_{u,1} = 1$ and $x_{v,2} = 1$ for nodes $u$ and $v$ indicates that node $u$ is in Cluster 1 and that node $v$ is in Cluster 2. Thus, the nodes are in different clusters. In this case, a value of 1 is contributed to the cost if and only if $c_{uv} = -1$.

Note that when using QAOA for maximum agreement correlation clustering, we need to take the negative of the above cost function, since the expected cost (5) is minimized.

## 4. Sub-Problem Quantum Alternating Operator Ansatz

In this section, we introduce the Sub-Problem Quantum Alternating Operator Ansatz, a generalization of QAOA that, as we will show in Section 5, leads to better results when solving the correlation clustering problem regarding both the approximation ratio and the used resources while maintaining the optimality guarantee for $p \rightarrow \infty$. In comparison to QAOA, we make two significant changes: Firstly, we employ nucleus sampling (Holtzman et al., 2020) for the computation of the expected cost. Secondly, we alter the ansatz itself by splitting the problem into sub-problems and applying QAOA to each of them.

**Nucleus Sampling** As is typical for variational algorithms, QAOA minimizes the expectation value of the classical cost function, which is represented by a cost Hamiltonian (5). Since the expectation value acts as an upper bound on the ground state energy, i.e., the optimal cost, this minimization approximates ground states of the cost Hamiltonian, and thus optimal solutions.

The upper diagram of Figure 2 shows the agreements of the basis states of the multi-level QAOA formulation of Weggemans et al. (2022) with $p = 1$ for a correlation clustering problem instance and the corresponding probabilities of these basis states. As expected, the algorithm shifts probability mass to solutions of low cost (i.e., high agreement).

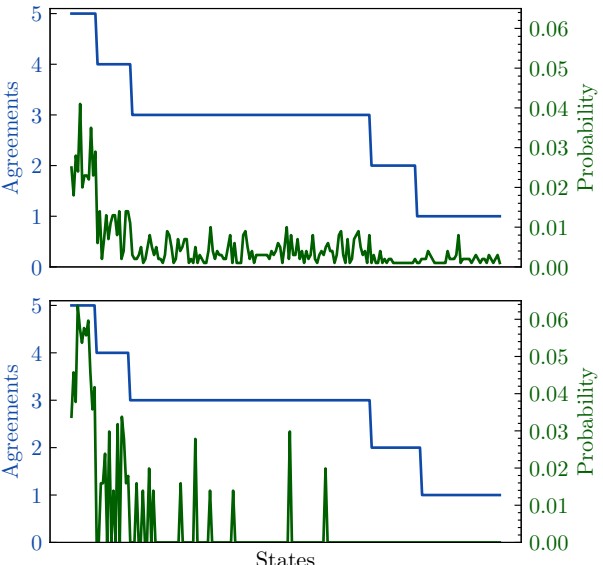

*Figure 2.* Depicted are two diagrams showing the probability of measuring basis states when applying the multi-level QAOA formulation of Weggemans et al. (2022) with $p = 1$ to the correlation clustering problem instance given in Figure 1. Shown in addition are the agreements of these basis states, i.e., the value of the cost function in (6). The probabilities of the diagram at the top are obtained directly from the QAOA results. The probabilities of the diagram at the bottom are obtained by nucleus sampling with a threshold of $t = 0.5$.

However, solutions of high cost are not completely erased and, due to their number, increase the expectation value significantly. The problem of this "unreliable tail" also occurs in decoding strategies for large language models and was approached by Holtzman et al. (2020) using a technique called nucleus sampling (sometimes called top-p sampling).

The main idea of nucleus sampling is that, instead of sampling directly from a given probability distribution, we sample from the most probable states whose cumulative probability surpasses a previously defined threshold. The set of those states is called the nucleus. Hence, in our case, we do not compute the expected cost with respect to the $n$-qubit state $|\psi\rangle$ obtained from the quantum algorithm, but with respect to $|\psi'\rangle$ obtained in the following way. Firstly, we set a threshold $t \in (0, 1]$ and compute a nucleus, i.e., a smallest set $X^{(t)} \subseteq \{0, 1\}^n$ such that

$$\sum_{x \in X^{(t)}} |\langle x \,|\, \psi \rangle|^2 \geq t \ .$$

Secondly, we set the probability amplitudes of the basis states not in $X^{(t)}$ to zero and use

$$t' = \sqrt{\sum_{x \in X^{(t)}} |\langle x \,|\, \psi \rangle|^2}$$

to rescale the remaining states accordingly, i.e., we set $|\psi'\rangle$ such that

$$\langle x \,|\, \psi' \rangle = \begin{cases} \langle x \,|\, \psi \rangle \,/\, t' & \text{if } x \in X^{(t)} \\ 0 & \text{otherwise} \end{cases} \ .$$

For the case $t = 1$, it holds $|\psi'\rangle = |\psi\rangle$, and our approach specializes to regular sampling. Moreover, since $|\psi'\rangle$ is also a normalized quantum state, the obtained cost function still acts as an upper bound on the ground state energy.

The lower diagram of Figure 2 shows the probabilities obtained when using multi-level QAOA with nucleus sampling and a threshold of $t = 0.5$ for training parameters and inference. Clearly, reducing the threshold from 1 leads to better solutions.

**Sub-Problems**   Instead of solving the whole correlation clustering problem at once by applying QAOA (1), we split it into $l$ sub-problems, solving each with QAOA, and introduce transition operators $U_{T_i}$ for $i \in \{1, \ldots, l\}$, preparing their initial state:

$$|\boldsymbol{\beta}, \boldsymbol{\gamma}\rangle^{\text{SQAOA}} := \prod_{i=1}^{l} \Big( \prod_{j=1}^{p} U_{M_i}(\boldsymbol{\beta_{i,j}}) U_{C_i}(\boldsymbol{\gamma_{i,j}}) \Big) U_{T_i} |\mathbf{0}\rangle \ . \tag{7}$$

Clearly, for $l = 1$, $U_{T_1} |\mathbf{0}\rangle = |s\rangle$ and operators $U_{M_1}$, $U_{C_1}$ satisfying properties (2-4), this specializes to QAOA, so we are again considering a proper generalization.

In order to apply this ansatz to an instance of the (unweighted maximum agreement) correlation clustering problem given by an undirected graph $G = (V, E)$ with $n = |V|$ nodes and costs $c \in \{+1, -1\}^E$, we need to choose the number of sub-problems $l$ and the corresponding operators appropriately. We do so by modelling correlation clustering as an iterated application of max-cut. In each of $l = n - 1$ iterations, we solve a max-cut problem restricted to those nodes that have not been assigned to a cluster in a previous iteration. The nodes that are labeled 0 by the solution to the max-cut problem are assigned to a new cluster. The nodes labeled 1 remain unassigned (or are assigned to a "final" cluster if it is the last iteration).

Realizing this directly within the framework of (7) is possible but requires complicated operators and, since at least one qubit per node is needed for the decision in each sub-problem, $\Omega(n^2)$ qubits which is worse than the approaches presented by Weggemans (2020) and Weggemans et al. (2022). However, since the qubits associated with a sub-problem are only manipulated by its operators, there is no interference between states that differ in one of those qubits once the sub-problem is processed. Consequently, we can measure all qubits of a sub-problem after applying its operators and evaluate the following sub-problem only on

the classical probability distribution estimated from these measurements instead of further performing operations on the whole quantum system. This fact allows for a more efficient implementation described in the following and in more detail in Algorithm 1.

We choose $l = n - 1$. For any sub-problem $i \in \{1, \dots, l\}$ and any solution $|x\rangle$ of the previous sub-problem, we compute

$$|\boldsymbol{\beta}, \boldsymbol{\gamma}\rangle_{i,x}^{\text{SQAOA}} := \Big( \prod_{j=1}^{p} U_{M_{i,x}}(\boldsymbol{\beta_{i,j}}) U_{C_{i,x}}(\boldsymbol{\gamma_{i,j}}) \Big) U_{T_{i,x}} |\mathbf{0}\rangle \,, \tag{8}$$

where

$$U_{T_{i,x}} = \bigotimes_{u \in V_{i,x}} H_u \,, \tag{9}$$

$$U_{M_{i,x}}(\beta) = e^{-i\beta \sum_{u \in V_{i,x}} X_u} \,, \tag{10}$$

$$U_{C_{i,x}}(\gamma_1, \gamma_2) = e^{-i\gamma_1 \sum_{uv \in E_{i,x}} c_{uv} Z_u Z_v} \tag{11}$$
$$e^{-i\gamma_2 \sum_{u \in V_{i,x}} w_u Z_u} \,,$$

$G_{i,x} = (V_{i,x}, E_{i,x})$ is the graph obtained from $G$ by removing nodes decided, i.e., labeled 0, in solution $|x\rangle$ of the previous sub-problem and $w \in \mathbb{R}^V$ are weights fulfilling $w_u^2 \neq w_v^2$ for all distinct nodes $u, v \in V$. For the first sub-problem we need to consider the whole graph $G$; therefore, we set $|x\rangle$ to $|\mathbf{1}\rangle$, i.e., all nodes are yet undecided. After preparing the state $|\boldsymbol{\beta}, \boldsymbol{\gamma}\rangle_{i,x}^{\text{SQAOA}}$, we estimate the corresponding probability distribution by sampling repeatedly from it and continue evaluating the next sub-problem on all states with non-zero probability. Once all sub-problems are processed, the expected costs are computed from the measured probability distributions. Figure 3 illustrates the described procedure compared to the multi-level approach of Weggemans et al. (2022).

The transition operator $U_{T_{i,x}}$ uses Hadamard gates to construct an equal superposition of all feasible states of the sub-problem. The mixing operator $U_{M_{i,x}}$ enables transitions between the feasible states of a sub-problem by flipping qubits, i.e., by changing if the corresponding nodes remain in the current cluster, or are assigned to a new cluster that is further split in the next sub-problem. The phase-separation operator $U_{C_{i,x}}$ incorporates the cost function into the first exponent, as described in Section 3, but drops constant terms since they affect all states in the same way and can be neglected. Additionally, the Hamiltonian given by the second exponent allows a cost-independent separation of phases based on individual nodes. The motivation behind introducing this second term with weights $w$, as well as the reason for restricting those, will be discussed in Section 5.

As described, we evaluate each sub-problem with respect to all solutions having a non-zero probability in the previous

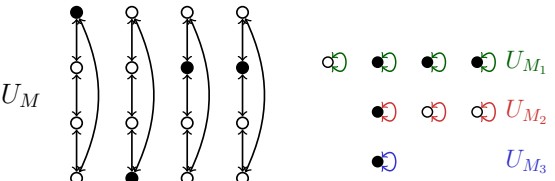

Figure 3. Depicted is are basis state of the multi-level QAOA formulation of Weggemans et al. (2022) (left) and SQAOA (right) corresponding to the same solution of a correlation clustering problem instance with four nodes. In the presented solution, the first node is assigned to the first cluster, the second node to the fourth cluster, and the remaining nodes to the second cluster. For QAOA, each column represents a qudit and each node a qudit state. For SQAOA, each node represents a qubit. Nodes colored black correspond to qudit states or qubits set to $|1\rangle$, and white nodes to qudit states or qubits set to $|0\rangle$. The arrows indicate transitions realized by the mixing operator. For SQAOA, the colors indicate the three sub-problems where, due to the measurements, the same physical qubits can be used for all sub-problems.

sub-problem. Since there can be exponentially many of these for dense probability distributions, this may constitute a performance bottleneck. However, since we apply nucleus sampling, as discussed previously, we only need to evaluate the next problem on the states in the nucleus. Although this does not guarantee a sub-exponential number of evaluations, we show experimentally in Section 5 that this number remains almost constant for the considered problem sizes and low nucleus sampling thresholds.

Note that, besides the sum over $V_{i,x}$ in $U_{C_{i,x}}$, the ansatz for a sub-problem corresponds to the one used for solving max-cut with the Quantum Approximate Optimization Algorithm (Farhi et al., 2014). Without this second term in the phase-separation operator, only pairwise interactions between nodes weighted by the costs would be considered for the phase-separation. Including it allows to also take individual nodes with weights $w$ into account. Note further that, while all other terms are permutation invariant, and we thus expect the parameters $\beta$ and $\gamma_1$ to be reusable across instances as for QAOA, the newly introduced term with parameter $\gamma_2$ is not, leading to the necessity of relearning it for different problem instances, or even the same instance when permuting nodes.

As shown by Weggemans (2020), operators (9-11) can be implemented using only Hadamard, RX, RZ and CX gates without requiring additional ancilla qubits. Moreover, since we measure after processing a sub-problem, qubits can be reused, and thus only a total of $n$ qubits are needed for the whole algorithm.

**Algorithm 1** SQAOA - Correlation Clustering

$probabilities =$ empty dictionary
$current\_states =$ empty set
add $\mathbf{1}$ to $current\_states$
**for** $i = 1$ **to** $l$ **do**
  $next\_states =$ empty set
  **for** $x$ **in** $current\_states$ **do**
    **for** $shot = 1$ **to** $1000$ **do**
      $|\psi\rangle = U_{T_{i,x}} |\mathbf{0}\rangle$
      **for** $j = 1$ **to** $p$ **do**
        $|\psi\rangle = U_{M_{i,x}}(\beta_{i,j})U_{C_{i,x}}(\gamma_{i,j}) |\psi\rangle$
      measure $|\psi\rangle$ and update $probabilities[i][x]$
    add nucleus of $probabilities[i][x]$ to $next\_states$
  $current\_states = next\_states$

$costs =$ empty dictionary
**for** $i = l$ **to** $1$ **do**
  **for** $y$ **in** $probabilities[i]$ **do**
    $costs[i][y] = 0$
    **for** $p, x$ **in** nucleus of $probabilities[i][y]$ **do**
      **for** $u, v$ **in** $E_{i,y}$ **do**
        **if** $c_{uv} == 1$ **then**
          $costs[i][y] \mathrel{+}= p\,(1 - x_u)\,(1 - x_v)$
          **if** $i == l$ **then**
            $costs[i][y] \mathrel{+}= p\,x_u\,x_v$
        **if** $c_{uv} == -1$ **then**
          $costs[i][y] \mathrel{+}= p\,(1 - x_u)\,x_v$
               $+\, p\,x_u\,(1 - x_v)$
        **if** $i \neq l$ **then**
          $costs[i][y] \mathrel{+}= p\,cost[i + 1][x]$

**return** $costs[1][\mathbf{1}]$

## 5. Evaluation

In this section, we evaluate our SQAOA formulation for correlation clustering. Firstly, we show that for $p \to \infty$, there exist parameters for each problem instance such that an optimal solution is obtained with certainty. Secondly, we experimentally compare our approach to the one of Weggemans et al. (2022) in terms of approximation ratios and runtimes.

**Theoretical Analysis** QAOA yields an optimal solution under the conditions given by Binkowski et al. (2024), containing especially $p \to \infty$ and that the optimal solution is an eigenvector of the phase-separation operator with the smallest eigenvalue. Although the given operators fulfill these conditions, this argument only guarantees to reach optimal solutions on the individual sub-problems, but not a globally optimal solution. Considering only pairwise interactions for the mixing operator, we have not been able to prove or disprove that there exist parameters such that (8) yields

a globally optimal solution for $p \to \infty$. However, when including the term for individual nodes, we have been able to show this by adapting the universality proof for the Quantum Approximate Optimization Algorithm from Morales et al. (2020) as shown in the following. The proofs of the upcoming lemmata are deferred to Appendix B.

**Definition 5.1.** Given a set of Hamiltonians $\mathcal{P} = \{H_1, H_2, \ldots, H_q\}$, we call the smallest real Lie algebra $\mathcal{L}$ with the commutator as the Lie bracket containing the elements of $\mathcal{P}$ the generated Lie algebra of $\mathcal{P}$.

**Proposition 5.2.** *(D'Alessandro, 2021) Let $\mathcal{P}$ be a set of Hamiltonians and let $\mathcal{L}$ be the generated Lie algebra of $\mathcal{P}$. The set of unitaries that can be approximated to arbitrary precision by iterated application of the elements in $\mathcal{P}$ is given by*

$$\{e^{-iA} \mid A \in \mathcal{L}\}.$$

**Lemma 5.3.** *Let $G = (V, E)$ be an undirected graph, let $c \in \mathbb{R}^E$ and $w \in \mathbb{R}^V$. Let further*

$$H_M = \sum_{u \in V} X_u, \quad H_C = \sum_{uv \in E} c_{uv} Z_u Z_v + \sum_{u \in V} w_u Z_u$$

*and let $\mathcal{L}$ be the generated Lie algebra of $\{H_M, H_C\}$. It holds that*

$$H_{C'} := \sum_{u \in V} w_u Z_u \in \mathcal{L}.$$

**Lemma 5.4.** *Let $G = (V, E)$ be an undirected graph and let $w \in \mathbb{R}^V$. Let further*

$$H_M = \sum_{u \in V} X_u, \quad H_{C'} = \sum_{u \in V} w_u Z_u$$

*and let $\mathcal{L}$ be the generated Lie algebra of $\{H_M, H_{C'}\}$. If $w_u{}^2 \neq w_v{}^2$ for all distinct $u, v \in V$, it holds for all $u' \in V$ that*

$$H_{u'}, X_{u'} \in \mathcal{L}.$$

**Theorem 5.5.** *Let $G = (V, E)$ be an undirected graph, let $c \in \mathbb{R}^E$ and $w \in \mathbb{R}^V$ with $w_u{}^2 \neq w_v{}^2$ for all distinct $u, v \in V$. Let further*

$$U_T = \bigotimes_{u \in V} H_u,$$

$$U_M(\beta) = e^{-i\beta \sum_{u \in V} X_u} \quad and$$

$$U_C(\gamma_1, \gamma_2) = e^{-i(\gamma_1 \sum_{uv \in E} c_{uv} Z_u Z_v + \gamma_2 \sum_{u \in V} w_u Z_u)}.$$

*For any basis state $|x\rangle$ with $x \in \{0, 1\}^{|V|}$, there exist parameters $\beta_j, \gamma_j \in \mathbb{R}$ for $j \in \{1, \ldots, p\}$ and a phase shift $\theta \in \mathbb{R}$ such that it holds for $p \to \infty$:*

$$e^{-i\theta} |x\rangle = \left(\prod_{j=1}^{p} U_M(\beta_j) U_C(\gamma_{1,j}, \gamma_{2,j})\right) U_T |\mathbf{0}\rangle.$$

*Proof.* Let $\mathcal{L}$ be the generated Lie algebra of $\{H_M, H_C\}$. After applying $U_T$, the qubits are in state $U_T |\mathbf{0}\rangle$. Since $H_u \in \mathcal{L}$ by Lemma 5.3 and Lemma 5.4, we can revert this state to $|\mathbf{0}\rangle$ (modulo a phase shift of $(-i)^{|V|}$) by applying $\prod_{u \in V} e^{-i\pi/2 H_u} = (-i)^{|V|} \bigotimes_{u \in V} H_u$. Next, we can (modulo a phase shift of $-i$) flip individual qubits associated with nodes $u \in V$ by applying $e^{-i\pi/2 X_u} = -iX_u$, since $X_u \in \mathcal{L}$ by Lemma 5.3 and Lemma 5.4. This allows to construct arbitrary basis states $|x\rangle$ (modulo a potential phase shift $e^{-i\theta}$). $\qquad\square$

According to Theorem 5.5, arbitrary basis states can be constructed in each sub-problem when $p$ approaches infinity. Therefore, for each instance of the correlation clustering problem, there clearly exist parameters such that our SQAOA formulation (8) obtains optimal solutions with certainty.

**Empirical Analysis**  To demonstrate the advancements of SQAOA and nucleus sampling in general, we conduct experiments on instances of the correlation clustering problem involving complete graphs and Erdös-Rényi graphs where the probability of an edge being present is $0.5$. We then compare these results with those of the multi-level QAOA formulation presented by Weggemans et al. (2022). The code for the SQAOA experiments is available at https://github.com/fabian-na/SQAOA.

For the experimental setup, we mainly follow Weggemans et al. (2022). In particular, for a fixed graph size, we evaluate the performance on datasets consisting of 50 problem instances with edge weights $\{+1, -1\}$, where the probability of an edge having weight $+1$ is uniformly increased from $0$ to $1$ to represent all weight configurations. Mean values and standard deviations given in this section always refer to the results obtained for a dataset, i.e., a mean approximation ratio of 1.0 with a standard deviation of 0.0 indicates that all 50 instances are solved to optimality.

For the classical optimization procedure, we use the Powell optimizer, which has proven to be efficient for solving other problems with QAOA (Pellow-Jarman et al., 2021; Fernández-Pendás et al., 2022). Further, for each dataset, we first learn parameters for the instance with all edges having weight $-1$ and then use these parameters as an initial point for the remaining instances in that dataset. The only exception are the parameters $\gamma_2$ used in the phase-separation operator of SQAOA. Those are, due to their permutation dependence, always initialized to $0$. The corresponding weights $w$ are chosen by enumerating all nodes by integers ranging from $1$ to $n$. With this choice, the phase-separation operator remains $2\pi$-periodic with respect to each of its parameters. We set the number of shots used to estimate probability distributions to 1000 and restart each optimization procedure 5 times, taking only the best overall result.

*Table 1.* Mean approximation ratios and runtimes for solving 50 correlation clustering problem instances on Erdős-Rényi Graph graphs with $n = 3, 4, 5$ nodes using the multi-level QAOA formulation of Weggemans et al. (2022) and SQAOA with a depth of $p = 1$ and thresholds for nucleus sampling of $t = 1, 0.1$.

Erdős-Rényi Graphs - Approximation Ratio

|  | $n = 3$ | $n = 4$ | $n = 5$ |
|---|---|---|---|
| QAOA $t = 1$ | $0.97 \pm 0.04$ | $0.92 \pm 0.07$ | $0.92 \pm 0.07$ |
| QAOA $t = 0.1$ | $1.00 \pm 0.00$ | $1.00 \pm 0.00$ | $1.00 \pm 0.00$ |
| SQAOA $t = 1$ | $0.94 \pm 0.08$ | $0.88 \pm 0.09$ | $0.85 \pm 0.10$ |
| SQAOA $t = 0.1$ | $1.00 \pm 0.00$ | $1.00 \pm 0.02$ | $1.00 \pm 0.02$ |

Erdős-Rényi Graphs - Runtime [s]

|  | $n = 3$ | $n = 4$ | $n = 5$ |
|---|---|---|---|
| QAOA $t = 1$ | $121 \pm 40$ | $221 \pm 30$ | $316 \pm 44$ |
| QAOA $t = 0.1$ | $5 \pm 0$ | $15 \pm 2$ | $109 \pm 32$ |
| SQAOA $t = 1$ | $23 \pm 12$ | $161 \pm 45$ | $799 \pm 256$ |
| SQAOA $t = 0.1$ | $2 \pm 1$ | $6 \pm 2$ | $10 \pm 5$ |

Table 1 shows approximation ratios and runtimes obtained for multi-level QAOA and SQAOA with depth $p = 1$ on correlation clustering instances of Erdős-Rényi graphs with $n = 3, 4, 5$ nodes and two thresholds $t = 1$ and $t = 0.1$ for nucleus sampling. An extended version of this table containing results for complete graphs, depths $p = 2, 3$ and threshold $t = 0.5$ is given in Appendix C. As can be seen from the table, both QAOA and SQAOA perform, even for $t = 1$, significantly better than the Swamy bound of 0.7666 (Swamy, 2004) with SQAOA achieving slightly worse approximation ratios. Setting the threshold to $t = 0.1$ greatly improves the approximation ratios, leading to optimal results for QAOA and near optimal results for SQAOA. While improving the approximation ratios, reducing the threshold also leads to an overall reduction of the runtime for the given experiments. However, this does not hold in general. As shown in the appendix, solving QAOA with $t = 0.5$ takes significantly longer than solving QAOA with $t = 1$ for the dataset with 5 nodes. For SQAOA, we do not observe such a behavior; in fact, the runtimes seem to scale much better than for the QAOA approach when using a low threshold for nucleus sampling.

Since the threshold alters only the computation of the cost function for QAOA and not the quantum algorithm itself, the difference in runtimes must be caused by an increased number of function evaluations during the optimization procedure. This might be due to the fact that discontinuities get introduced to the cost function when states enter or leave the nucleus. For SQAOA on the other hand, reducing $t$ has always resulted in lower runtimes for the experiments we

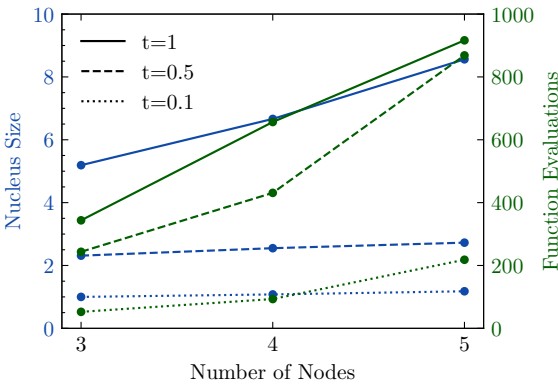

*Figure 4.* Depicted is a diagram showing the mean nucleus size and the mean number of function evaluations when solving the dataset of Erdős-Rényi graphs with 5 nodes using SQAOA with depth $p = 1$ for nucleus sampling thresholds of $t = 1, 0.5, 0.1$.

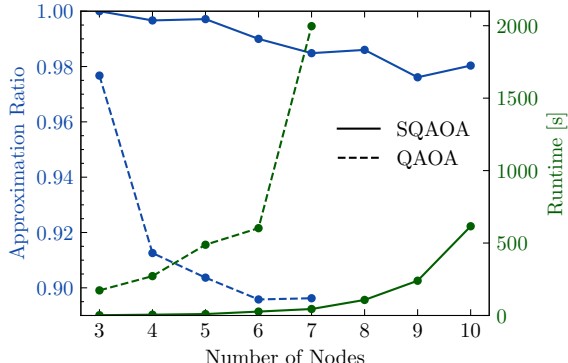

*Figure 5.* Depicted is a diagram showing the mean approximation ratios and runtimes of SQAOA ($t = 0.1$) and the multi-level QAOA approach of Weggemans et al. (2022) ($t = 1$) with ansatz depth $p = 1$ when applied to datasets of Erdős-Rényi graphs with up to 10 nodes for SQAOA and 7 nodes for QAOA.

have conducted. This is due to the fact that, even as the number of function evaluations increases, each evaluation takes less time since fewer elements are in the nucleus. This is illustrated in Figure 4 for the dataset of Erdős-Rényi graphs with 5 nodes and $p = 1$. In particular, one can see that the number of elements in the nucleus is almost constant for the considered problem sizes and low thresholds.

Weggemans et al. (2022) consider instances of the correlation clustering problem with up to 7 nodes. In Figure 5, we give approximation ratios and runtimes for SQAOA with $t = 0.1$ and $p = 1$ on instances with up to 10 nodes. As can be seen from the figure, the runtime increases exponentially, as expected, although slower than for the QAOA formulation. The approximation ratio, however, seems to remain almost constant, further corroborating the potential of SQAOA and variational algorithms in general.

## 6. Conclusion

We introduce the Sub-Problem Quantum Alternating Operator Ansatz (SQAOA), a generalization of the Quantum Alternating Operator Ansatz (QAOA) based on nucleus sampling and splitting problems into sub-problems. In a theoretical analysis, we show that for each instance of the correlation clustering problem, there exist parameters such that a specific SQAOA formulation of the problem obtains an optimal solution with certainty. Further, we show experimentally that this SQAOA formulation outperforms existing QAOA approaches for correlation clustering in terms of approximation ratios and runtime while using only as many qubits as there are elements to cluster.

We see two possible directions for future research: Further analyzing SQAOA for correlation clustering and extending its application to other problems. Regarding the first direction, we have not yet given a lower bound on the achieved approximation ratio, as it is done by Weggemans (2020) for the multi-level formulation. One could also consider modelling correlation clustering with different sub-problems since we do not exploit the full expressiveness of SQAOA with the current formulation, which uses the same operators for each sub-problem. Regarding the second direction, splitting a problem into sub-problems is a universal approach, and similar improvements may be possible for problems beyond correlation clustering. Of particular interest are thereby problems in which elements are assigned one of multiple labels. For example, one could consider the Maximum $k$-Colorable Subgraph Problem with sub-problems coloring parts of the graph that have not yet been considered using a fixed number of colors smaller than $k$.

## Acknowledgements

We thank Jordi Weggemans for providing the source code of Weggemans et al. (2022), which we use to perform the QAOA experiments. This work is partly supported by the Federal Ministry of Education and Research of Germany through DAAD Project 57616814 (SECAI) and Project 16KIS2332K (AI.Auto-Immune).

## Impact Statement

This theoretical article presents work whose goal is to advance the field of machine learning, more specifically clustering. As for all advances in this field, there are many potential societal consequences of our work. However, we do not feel that the implications of this article differ from those of other contributions to that field and must be specifically highlighted here.

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

# A. Quantum Gates

**Pauli-X:**
$$X := \begin{bmatrix} 0 & 1 \\ 1 & 0 \end{bmatrix} \quad X\,|0\rangle = |1\rangle$$

**Rotational X:**
$$RX(\theta) := e^{-i\theta X} = \cos(\theta)I - i\sin(\theta)X$$

**Pauli-Z:**
$$Z := \begin{bmatrix} 1 & 0 \\ 0 & -1 \end{bmatrix} \quad Z\,|1\rangle = -\,|1\rangle$$

**Rotational Z:**
$$RZ(\theta) := e^{-i\theta Z} = \cos(\theta)I - i\sin(\theta)Z$$

**Hadamard:**
$$H := \frac{1}{\sqrt{2}}\begin{bmatrix} 1 & 1 \\ 1 & -1 \end{bmatrix} \quad H\,|0\rangle = \frac{1}{\sqrt{2}}(|0\rangle + |1\rangle)$$

**Conditional X:**
$$CX := \begin{bmatrix} 1 & 0 & 0 & 0 \\ 0 & 1 & 0 & 0 \\ 0 & 0 & 0 & 1 \\ 0 & 0 & 1 & 0 \end{bmatrix} \begin{matrix} |00\rangle \\ |01\rangle \\ |10\rangle \\ |11\rangle \end{matrix}$$

# B. Additional Proofs

*Proof of Lemma 5.3.* We want to show that $H_{C'} = \sum_{u \in V} w_u Z_u$ is in the generated lie algebra $\mathcal{L}$ of $\{H_M, H_C\}$, where $H_M = \sum_{u \in V} X_u$ and $H_C = \sum_{uv \in E} c_{uv} Z_u Z_v + \sum_{u \in V} w_u Z_u$. For notational convenience, we define $H_{C_1} := \sum_{uv \in E} c_{uv} Z_u Z_v$.

In analogy to Morales et al. (2020), we define a series of commutators in $\mathcal{L}$, showing finally that $H_{C'} \in \mathcal{L}$:

$$
\begin{aligned}
H_{YZ} := \frac{1}{2i}[H_C, H_M] &= \frac{1}{2i}\big([H_{C_1}, H_M] + [H_{C'}, H_M]\big) \\
&= \sum_{uv \in E} c_{uv}(Z_u Y_v + Y_u Z_v) + \sum_{u \in V} w_u Y_u \in \mathcal{L} \ ,
\end{aligned}
$$

$$
\begin{aligned}
\frac{1}{2i}[H_{YZ}, H_M] &= \sum_{uv \in E} c_{uv}\Big[Z_u Y_v + Y_u Z_v, \sum_{u' \in V} X_{u'}\Big] + \sum_{u \in V} w_u\Big[Y_u, \sum_{u' \in V} X_{u'}\Big] \\
&= \sum_{uv \in E} c_{uv}\big(Y_u Y_v - Z_u Z_v - Z_u Z_v + Y_u Y_v\big) - \sum_{u \in V} w_u Z_u \\
&= 2\sum_{uv \in E} c_{uv}\big(Y_u Y_v - Z_u Z_v\big) - \sum_{u \in V} w_u Z_u \in \mathcal{L} \ ,
\end{aligned}
$$

$$
H_{(1)} := \frac{1}{2i}[H_{YZ}, H_M] + H_C = \sum_{uv \in E} c_{uv}\big(2Y_u Y_v - Z_u Z_v\big) \in \mathcal{L} \ ,
$$

$$
\begin{aligned}
H_{(2)} := \frac{1}{2i}[H_{(1)}, H_M] = \frac{1}{2i}[H_{(1)}, H_M] &= \frac{1}{2i}\sum_{uv \in E} c_{uv}\Big(2\Big[Y_u Y_v, \sum_{u' \in V} X_{u'}\Big] - \Big[Z_u Z_v, \sum_{u' \in V} X_{u'}\Big]\Big) \\
&= \sum_{uv \in E} c_{uv}\big(2\big(-Z_u Y_v - Y_u Z_v\big) - \big(Y_u Z_v + Z_u Y_v\big)\big) \\
&= -3\sum_{uv \in E} c_{uv}\big(Z_u Y_v + Y_u Z_v\big) \in \mathcal{L} \ ,
\end{aligned}
$$

$$
\begin{aligned}
\frac{1}{2i}[H_{YZ} + \frac{1}{3}H_{(2)}, H_M] &= \frac{1}{2i}\big([H_{YZ}, H_M] + [\frac{1}{3}H_{(2)}, H_M]\big) \\
&= -\sum_{u \in V} w_u Z_u + 2\sum_{uv \in E} c_{uv}(Y_u Y_v - Z_u Z_v) \\
&\quad - \sum_{uv \in E} c_{uv}\big(Y_u Y_v - Z_u Z_v - Z_v Z_u + Y_v Y_u\big) \\
&= -\sum_{u \in V} w_u Z_u \\
&= -H_{C'} \in \mathcal{L} \ .
\end{aligned}
$$

It follows directly from $-H_{C'} \in \mathcal{L}$ that $H_{C'} \in \mathcal{L}$.

$\square$

*Proof of Lemma 5.4.* We want to show that $H_{u'}$ and $X_{u'}$ are for all $u' \in V$ in the generated lie algebra $\mathcal{L}$ of $\{H_M, H_{C'}\}$, where $H_M = \sum_{u \in V} X_u$ and $H_{C'} = \sum_{u \in V} w_u Z_u$ with $w_u{}^2 \neq w_v{}^2$ for all $u, v \in V$.

Assume we have already shown $X_{u'} \in \mathcal{L}$. It follows directly that $Y_{u'} = \frac{1}{2i}[\frac{1}{w_{u'}} H_{C'}, X_{u'}] \in \mathcal{L}$, further $Z_{u'} = \frac{1}{2i}[X_{u'} Y_{u'}] \in \mathcal{L}$ and thus $H_{u'} = \frac{1}{\sqrt{2}}(Z + X) \in \mathcal{L}$. Consequently, it only remains to show $X_{u'} \in \mathcal{L}$.

Define $n = |V|$. For proving $X_{u'} \in \mathcal{L}$, we first show that if $H_{M'} = \sum_{u \in V'} w'_u X_u \in \mathcal{L}$ with $V' \subseteq V$, and $w'_u{}^2 \neq w'_v{}^2$ for all $u, v \in V'$, we can for any $x \in V'$ construct $\sum_{u \in V' \setminus \{x\}} w''_u X_u \in \mathcal{L}$ such that $w''_u{}^2 \neq w''_v{}^2$ for all $u, v \in V' \setminus \{x\}$.

In particular, it follows from

$$H_{Y'} := \frac{1}{2i}[H_C, H_{M'}] = \sum_{u \in V'} w_u w'_u{}^2 Y_u \in \mathcal{L}$$

and

$$H_{X'} := \frac{1}{2i}[H_{Y'}, H_C] = \sum_{u \in V^{(i)}} w_u^2 w'_u{}^2 X_u \in \mathcal{L},$$

that it holds for every $x \in V'$ that

$$w_x^2 H_{M'} - H_{X'} = \sum_{u \in V' \setminus \{x\}} (w_x^2 - w_u^2) w'_u{}^2 X_u \in \mathcal{L}.$$

Setting $w''_u{}^2 = (w_x^2 - w_u^2) w'_u{}^2$ yields the desired result.

It only remains to show that we can initially construct such an $H_{M'}$. Therefore, consider first

$$H_Y := \frac{1}{2i}[H_C, H_M] = \sum_{u \in V} w_u Y_u .$$

We then get

$$\frac{1}{2i}[H_Y, H_C] = \sum_{u \in V} w_u^2 X_u = H_{M'}$$

with $V' = V$ and $w' = w$. $\qquad\square$

# C. Additional Tables

*Table 2.* Mean approximation ratios and runtimes with standard deviations for solving 50 correlation clustering problem instances on complete and Erdős-Rényi graphs with $n = 3, 4, 5$ nodes using the multi-level QAOA formulation of Weggemans et al. (2022) and SQAOA, ansatz depths of $p = 1, 2, 3$ and thresholds for nucleus sampling of $t = 1, 0.5, 0.1$.

### Complete Graphs - Approximation Ratio

| | QAOA $t = 1$ | QAOA $t = 0.5$ | QAOA $t = 0.1$ | SQAOA $t = 1$ | SQAOA $t = 0.5$ | SQAOA $t = 0.1$ |
|---|---|---|---|---|---|---|
| | $0.97 \pm 0.04$ | $1.00 \pm 0.00$ | $1.00 \pm 0.00$ | $0.96 \pm 0.05$ | $1.00 \pm 0.00$ | $1.00 \pm 0.00$ |
| $n = 3$ | $1.00 \pm 0.00$ | $1.00 \pm 0.00$ | $1.00 \pm 0.00$ | $0.97 \pm 0.05$ | $1.00 \pm 0.00$ | $1.00 \pm 0.00$ |
| | | $1.00 \pm 0.00$ | $1.00 \pm 0.00$ | | $1.00 \pm 0.00$ | $1.00 \pm 0.00$ |
| | $0.91 \pm 0.07$ | $0.99 \pm 0.02$ | $1.00 \pm 0.00$ | $0.83 \pm 0.06$ | $0.98 \pm 0.02$ | $1.00 \pm 0.02$ |
| $n = 4$ | $0.98 \pm 0.02$ | $1.00 \pm 0.00$ | $1.00 \pm 0.00$ | $0.83 \pm 0.06$ | $1.00 \pm 0.00$ | $1.00 \pm 0.00$ |
| | | $1.00 \pm 0.00$ | $1.00 \pm 0.00$ | | $0.99 \pm 0.03$ | $0.99 \pm 0.04$ |
| | $0.90 \pm 0.08$ | $0.97 \pm 0.04$ | $0.98 \pm 0.04$ | $0.86 \pm 0.07$ | $0.97 \pm 0.03$ | $0.99 \pm 0.03$ |
| $n = 5$ | $0.95 \pm 0.00$ | $0.98 \pm 0.04$ | $0.98 \pm 0.05$ | $0.86 \pm 0.08$ | $0.98 \pm 0.03$ | $0.99 \pm 0.03$ |
| | | $0.98 \pm 0.04$ | $0.98 \pm 0.05$ | | $0.99 \pm 0.03$ | $0.99 \pm 0.03$ |

### Complete Graphs - Runtime [s]

| | QAOA $t = 1$ | QAOA $t = 0.5$ | QAOA $t = 0.1$ | SQAOA $t = 1$ | SQAOA $t = 0.5$ | SQAOA $t = 0.1$ |
|---|---|---|---|---|---|---|
| | $173 \pm 33$ | $6 \pm 1$ | $5 \pm 0$ | $44 \pm 9$ | $14 \pm 5$ | $2 \pm 1$ |
| $n = 3$ | $315 \pm 59$ | $16 \pm 2$ | $15 \pm 2$ | $100 \pm 21$ | $15 \pm 7$ | $4 \pm 2$ |
| | | $30 \pm 4$ | $21 \pm 5$ | | $27 \pm 1$ | $7 \pm 4$ |
| | $272 \pm 23$ | $40 \pm 8$ | $15 \pm 2$ | $207 \pm 50$ | $41 \pm 15$ | $9 \pm 2$ |
| $n = 4$ | $691 \pm 58$ | $85 \pm 17$ | $35 \pm 7$ | $422 \pm 77$ | $78 \pm 27$ | $15 \pm 4$ |
| | | $102 \pm 36$ | $55 \pm 16$ | | $85 \pm 22$ | $18 \pm 4$ |
| | $488 \pm 60$ | $716 \pm 302$ | $109 \pm 32$ | $1268 \pm 450$ | $155 \pm 78$ | $15 \pm 4$ |
| $n = 5$ | $1012 \pm 101$ | $1140 \pm 190$ | $216 \pm 35$ | $2438 \pm 836$ | $265 \pm 118$ | $26 \pm 9$ |
| | | $2361 \pm 641$ | $349 \pm 73$ | | $384 \pm 175$ | $50 \pm 15$ |

### Erdős-Rényi Graphs - Approximation Ratio

| | QAOA $t = 1$ | QAOA $t = 0.5$ | QAOA $t = 0.1$ | SQAOA $t = 1$ | SQAOA $t = 0.5$ | SQAOA $t = 0.1$ |
|---|---|---|---|---|---|---|
| | $0.97 \pm 0.04$ | $1.00 \pm 0.00$ | $1.00 \pm 0.00$ | $0.94 \pm 0.08$ | $1.00 \pm 0.00$ | $1.00 \pm 0.00$ |
| $n = 3$ | $1.00 \pm 0.01$ | $1.00 \pm 0.00$ | $1.00 \pm 0.00$ | $0.94 \pm 0.08$ | $1.00 \pm 0.00$ | $1.00 \pm 0.00$ |
| | | $1.00 \pm 0.00$ | $1.00 \pm 0.00$ | | $1.00 \pm 0.00$ | $1.00 \pm 0.00$ |
| | $0.92 \pm 0.07$ | $1.00 \pm 0.01$ | $1.00 \pm 0.00$ | $0.88 \pm 0.09$ | $0.99 \pm 0.03$ | $1.00 \pm 0.02$ |
| $n = 4$ | $0.97 \pm 0.03$ | $1.00 \pm 0.00$ | $1.00 \pm 0.00$ | $0.88 \pm 0.09$ | $1.00 \pm 0.02$ | $1.00 \pm 0.00$ |
| | | $1.00 \pm 0.00$ | $1.00 \pm 0.00$ | | $1.00 \pm 0.00$ | $1.00 \pm 0.00$ |
| | $0.92 \pm 0.07$ | $0.98 \pm 0.03$ | $1.00 \pm 0.00$ | $0.85 \pm 0.10$ | $0.98 \pm 0.03$ | $1.00 \pm 0.02$ |
| $n = 5$ | $0.96 \pm 0.04$ | $1.00 \pm 0.01$ | $1.00 \pm 0.00$ | $0.85 \pm 0.10$ | $0.99 \pm 0.03$ | $1.00 \pm 0.00$ |
| | | $1.00 \pm 0.01$ | $1.00 \pm 0.00$ | | $1.00 \pm 0.01$ | $1.00 \pm 0.00$ |

*Table 2.* (Continuation)

Erdős-Rényi Graphs - Runtime [s]

|  | QAOA $t = 1$ | QAOA $t = 0.5$ | QAOA $t = 0.1$ | SQAOA $t = 1$ | SQAOA $t = 0.5$ | SQAOA $t = 0.1$ |
|---|---|---|---|---|---|---|
|  | $121 \pm 40$ | $6 \pm 1$ | $5 \pm 0$ | $23 \pm 12$ | $8 \pm 4$ | $2 \pm 1$ |
| $n = 3$ | $272 \pm 84$ | $12 \pm 2$ | $15 \pm 2$ | $53 \pm 27$ | $15 \pm 8$ | $4 \pm 2$ |
|  |  | $26 \pm 3$ | $21 \pm 5$ |  | $23 \pm 12$ | $7 \pm 3$ |
|  | $221 \pm 30$ | $38 \pm 9$ | $15 \pm 2$ | $161 \pm 45$ | $38 \pm 18$ | $6 \pm 2$ |
| $n = 4$ | $501 \pm 98$ | $54 \pm 9$ | $35 \pm 7$ | $338 \pm 102$ | $52 \pm 25$ | $12 \pm 4$ |
|  |  | $139 \pm 26$ | $55 \pm 16$ |  | $97 \pm 37$ | $19 \pm 7$ |
|  | $316 \pm 44$ | $809 \pm 148$ | $109 \pm 32$ | $799 \pm 256$ | $127 \pm 61$ | $10 \pm 5$ |
| $n = 5$ | $748 \pm 125$ | $1285 \pm 304$ | $216 \pm 35$ | $2155 \pm 789$ | $193 \pm 94$ | $22 \pm 7$ |
|  |  | $1860 \pm 386$ | $349 \pm 73$ |  | $256 \pm 116$ | $28 \pm 9$ |

