# OpenReview forum: "A Sub-Problem Quantum Alternating Operator Ansatz for Correlation Clustering"
_ICML.cc/2025/Conference — ICML 2025 poster_

### Official Review · Reviewer_yi8D · 2025-03-12

**Overall Recommendation:** 3

**Summary:**

This paper proposed the QAOA for correlation clustering, by introducing a Sub Problem QAOA. This is motivated by the nucleus sampling and sub problem is dependent to solve correlation clustering. Although QAOA for correlation clustering has been studied in the literature, Weggemans et al. (2022) is restricted to 4-level qudits such that only solutions involving at most 4 clusters can be considered.

The paper guarantees that there exist parameters such that for depth $p \rightarrow \infty$  an optimal solution is obtained with certainty. The experiments are conducted in complete ad Erdos-Renyi graphs with 10 nodes.

**Claims And Evidence:**

Yes

**Essential References Not Discussed:**

N/A

**Experimental Designs Or Analyses:**

The comparison with multi-level QAOA formulation of Weggemans et al. (2022) was performed in simple setting,complete and Erdos-Renyi graphs where the probability of an
edge being present is 0.5. SQAOA outperforms existing QAOA approaches in terms of approximation ratios and runtime for $p=1$ up to 10 nodes.

**Methods And Evaluation Criteria:**

Among the correlation clustering formulation, the paper focuses on the unweighted maximum agreement correlation clustering on general graphs, which is APX-hard but admits a constant-ratio approximation.

Compared with QAOA, SPQAOA uses nucleus sampling for computing the cost function.
The set of states called nucleus is selected by the probability mass with threshold $t$ and $t=1$ recovers the standard sampling. $n-1$ subproblems are adapted to QAOA. Once all sub-problems are processed, the expected costs can be computed from the measured probability distributions

**Other Comments Or Suggestions:**

N/A

**Other Strengths And Weaknesses:**

The paper was very clear to read the details.
Introduction of Quantum Alternating Operator Ansatz was done by the elegant description even for non-professionals on quantum information.

**Questions For Authors:**

If we consider the minimum disagreement formulation, will  we need a different techniques of QAOA considered in Max-Agree?


Can we perform the case with larger depth $p$? Do we expect the performance becomes better? Is it prohibited due to computational resources?

**Relation To Broader Scientific Literature:**

The proposed method is improved over QAOA formulation of Weggemans et al. (2022).

SPQAOA is specific to correlation clustering, but it would be great to see the possibility of other applications.

**Theoretical Claims:**

The proof is based on (Morales, 2020) and (D’Alessandro, 2021). It seems that it is not very hard to guarantee the optimality for $p \rightarrow \infty$.

---

> ### Author Rebuttal · Authors · 2025-03-31
>
> We thank all reviewers for their constructive feedback.
> There seems to be consensus that our experimental and theoretical claims are correct and that the proposed approach surpasses previous QAOA methods for correlation clustering. Here, we gladly address remaining questions and concerns of Reviewer yi8D:
>
> **Applications beyond Correlation Clustering**
>
> The reviewer is concerned that we apply our approach only to correlation clustering.
> We agree that our article focuses on correlation clustering and have stated clearly in the title and abstract that it is about a generalization of QAOA for correlation clustering.
> Given the wide range of applications of correlation clustering and numerous papers, especially in recent years, focusing on this problem specifically, we think that this problem is interesting enough to be considered on its own.
> Furthermore, the constraints have made it particularly hard to apply QAOA to correlation clustering in the past, allowing us to demonstrate the effectiveness of our proposed generalization.
>
> Beyond correlation clustering, candidates for our sub-problem approach are ones in which elements are assigned one of multiple labels. E.g., one could consider the Maximum $k$-Colorable Subgraph Problem with sub-problems coloring so-far-unconsidered parts of the graph by a fixed number of colors smaller than $k$.
>
> **Comparison with Wegemanns et al.**
>
> The reviewer is concerned that we compare our work only with that of Wegemanns et al. 2022.
> We emphasize that the work of Wegemanns et al. is the only work that applies QAOA to the correlation clustering problem. In an article from 2020 different QAOA formulations are analyzed, the multi-level formulation is found to perform best and is further evaluated in an article from 2022. We have chosen to compare our approach with this best-performing method because it is the state of the art.
>
> **Minimum Disagreement Formulation**
>
> The approach is easily adapted to the minimum disagreement formulation (and to other linear objective functions with the same set of constraints) by modifying the phase-separation operator.
> Since the objective value of the minimum disagreement formulation differs from that of the maximum agreement formulation only by an additive constant, one can even use the same phase-separation operator in this particular case and add the constant in classical post-processing.
>
> **Performance with Larger Ansatz Depth $\mathbf{p}$**
>
> The reviewer is right, performance does increase with larger $p$, but using it is prohibited by computational resources. To make this explicit, we suggest adding the following paragraph after Line 154.
>
> "The approximation ratios are expected to increase with larger ansatz depth $p$ and are guaranteed to improve with optimal parameters.
> However, the depth is limited firstly, because the number of applied operators increases, resulting in problems with computational resources for the simulation on classical computers and the introduction of noise for the execution on quantum computers, and, secondly, to a lesser extent, since the number of learnable parameters increases with $p$ and gradients cannot be computed easily for the quantum circuit."

---

### Official Review · Reviewer_L5UV · 2025-03-14

**Overall Recommendation:** 3

**Summary:**

The paper constructs a new variant of QAOA (rather) specifically for the correlation clustering problem and shows improved performance over QAOA.

**Claims And Evidence:**

Yes.

**Essential References Not Discussed:**

none

**Experimental Designs Or Analyses:**

Experimental analysis is done correctly, although very narrow in what is considered. This refers to both focusing only on a very specific problem domain and also to only considering the one chosen approach. Crucially, empirical evaluation lacks a proper ablation study to find out what parts of the approach really contribute to the results.

**Methods And Evaluation Criteria:**

Yes.

**Other Comments Or Suggestions:**

Introduction is way too long. Basically, the first paragraph can be cut directly.

mathcal{S} is weirdly introduced in line 163

"there is no interference" is repeated twice (lines 262 -- 265)

**Other Strengths And Weaknesses:**

It is clear that the chosen approach is quite fitted to the problem of correlation clustering. However, potential for other or similarly structured problems should be discussed if not even evaluated.

**Questions For Authors:**

none

**Relation To Broader Scientific Literature:**

Generalizing QAOA in the way it is described here appears powerful. However, focusing this strongly on correlation clustering severely limits any potential impact.

**Theoretical Claims:**

At a quick look, the proof seems fine. The proven property also is simple enough.

---

> ### Author Rebuttal · Authors · 2025-03-31
>
> We thank all reviewers for their constructive feedback. There seems to be consensus that our experimental and theoretical claims are correct and that the proposed approach surpasses previous QAOA methods for correlation clustering. Here, we gladly address remaining questions and concerns of Reviewer L5UV:
>
> **Applications beyond Correlation Clustering**
>
> The reviewer is concerned that we apply our approach only to correlation clustering. We agree that our article focuses on correlation clustering and have stated clearly in the title and abstract that it is about a generalization of QAOA for correlation clustering. Given the wide range of applications of correlation clustering and numerous papers, especially in recent years, focusing on this problem specifically, we think that this problem is interesting enough to be considered on its own. Furthermore, the constraints have made it particularly hard to apply QAOA to correlation clustering in the past, allowing us to demonstrate the effectiveness of our proposed generalization.
>
> Beyond correlation clustering, candidates for our sub-problem approach are ones in which elements are assigned one of multiple labels. E.g., one could consider the Maximum $k$-Colorable Subgraph Problem with sub-problems coloring so-far-unconsidered parts of the graph by a fixed number of colors smaller than $k$.
>
> **Comparison with Wegemanns et al.**
>
> The reviewer is concerned that we compare our work only with that of Wegemanns et al. 2022.
> We emphasize that the work of Wegemanns et al. is the only work that applies QAOA to the correlation clustering problem. In an article from 2020 different QAOA formulations are analyzed, the multi-level formulation is found to perform best and is further evaluated in an article from 2022. We have chosen to compare our approach with this best-performing method because it is the state of the art.
>
>
> **Ablation Study**
>
> The reviewer notes that our empirical evaluation lacks an ablation study.
> The presented approach consists of two main parts, nucleus sampling and the splitting of the given problem into sub-problems.
> We evaluate both independently and in combination, in Table 1 and Appendix C, showing that nucleus sampling improves the approximation ratio while splitting into sub-problems reduces the runtime.
> We agree with the reviewer that especially the splitting could be analyzed further.
> We have in fact done this (not shown in the article) for a preliminary version of SQAOA that also parametrizes the transition operators.
> We have found that this parametrization does not improve the approximation ratio, so we have decided not to report this negative result. Of course, we are prepared to do so in the supplement, should the reviewer recommend it.
>
> **Additional, Minor Corrections**
>
> We understand the reviewer's comment that the introduction is rather long. To strike a balance between conciseness and accessibility, we propose to remove the sentence "For example, Shor’s factoring algorithm (Shor, 1997) provides an exponential speed-up over the best-known classical factoring algorithm." from the first paragraph. We are
> prepared to shorten the introduction further if there is consensus among the reviewers that we should do so.
>
> To introduce $\mathcal{S}$ more clearly, we propose to replace Line 161 by "$|s\rangle$ is an initial state in the feasible space $\mathcal{S}$, which is given by the set of all superpositions of classically feasible states, i.e. by".
>
> We have removed "there is no interference between those states, and" in Line 264.

---

### Official Review · Reviewer_JUTn · 2025-03-14

**Overall Recommendation:** 2

**Summary:**

The paper presents a new quantum optimization approach called the Sub-Problem Quantum Alternating Operator Ansatz (SQAOA) aimed at solving correlation clustering problems. The approach modifies the Quantum Alternating Operator Ansatz (QAOA) by introducing two key modifications: 1) it uses nucleus sampling to compute the cost function and 2) it divides the problem into sub-problems, solving each one individually with QAOA. The authors argue that these changes lead to a method that is more suitable for the specific challenges posed by correlation clustering. Theoretical guarantees are provided, showing that SQAOA can obtain optimal solutions as the depth of the quantum circuit approaches infinity (p → ∞). Experimental results demonstrate that SQAOA outperforms standard QAOA in terms of approximation ratios and runtimes on various graphs, such as complete and Erdős-Rényi graphs.

**Claims And Evidence:**

The main contributions of this paper is clear and well supported. However, the claim that QAOA (quantum alternating operator ansatz) is considered a promising candidate for achieving quantum supremacy is not convincing to me. There are obviously other issues such as the number of shots to achieve desired accuracy, gradient update when $p\rightarrow \infty$, and etc. Without discussing all the aspects, it is not convincing to claim that QAOA is a promising candidate for quantum supremacy.

**Essential References Not Discussed:**

No

**Experimental Designs Or Analyses:**

With only one baseline method and most of the experiments are conducted on graphs with less than 6 nodes, the experiments can only illustrate that the proposed method surpass Weggemans et al., 2022.

**Methods And Evaluation Criteria:**

First of all, there is a mixup in quantum alternating operator ansatz and quantum approximation optimization algorithm, and the abbreviation QAOA requires a more specific assignment to which one of the above (or the multilevel QAOA in Weggemans et al., 2022). Besides, there are a few other issues.
1. The correlation clustering (CC) problem requires more explanation. As the only testbed for the proposed algorithm, we need more information including how the authors convert the classical cost function to Hamiltonian, etc. The authors have also to convince the readers why we should focus on this specific type of Combinatorial Optimization problems instead of targeting general COs. The whole paper depends so much on a previous one (Weggemans et al., 2022).  I've quickly gone through this reference paper and I believe the reason why (Weggemans et al., 2022) choose to solve CC problem is because they would like to demonstrate the design of qudit instead of qubit on neutral atom quantum computers. And qudit is a perfect fit for clustering multiple labels. However, in this paper the authors are trying to translate the algorithm back to qubits, which seems not a good motivation.
2. The definitions of Aggreements and Probability in Figure 1 are vague and need further clarification. The introduction of nucleus sampling also introduced a hyperparameter $t$, which seems deciding the value of $t$ is non-trivial and will be hard to determine with larger problem scales.

**Other Comments Or Suggestions:**

No

**Other Strengths And Weaknesses:**

No

**Questions For Authors:**

No

**Relation To Broader Scientific Literature:**

The authors translate a previous literature for correlation clustering problem with qudits to qubits and proposed a sampling method to accelerate the convergence.

**Theoretical Claims:**

The theoretical claims are legit, but the main problem is it is not surprise for me. The authors can refer to [1] for the details of overparameterization theorem and I think as long as the alternating layers do not commute with each other and generate enough Lie Algebra dimension, they are guaranteed to achieve the desired state.

[1] Larocca, Martin, et al. "Theory of overparametrization in quantum neural networks." Nature Computational Science 3.6 (2023): 542-551.

---

> ### Author Rebuttal · Authors · 2025-03-31
>
> We thank all reviewers for their constructive feedback.
> There seems to be consensus that our experimental and theoretical claims are correct and that the proposed approach surpasses previous QAOA methods for correlation clustering. Here, we gladly address remaining questions and concerns of Reviewer JUTn:
>
> **Quantum Supremacy**
>
> We agree that the presented approach does not solve existing problems of QAOA for achieving quantum supremacy, such as the number of shots or the gradient update for increasing ansatz depths, and we do not intend to make this claim.
> We also see that the introductory statement "QAOA is considered a promising candidate for achieving quantum supremacy for the following reasons:" does not account for the problems with QAOA that need yet to be solved, and propose to replace it by "QAOA is considered a promising variational quantum algorithm for the following reasons:".
>
> **QAOA**
>
> The Quantum Alternating Operator Ansatz is a generalization that was established after the Quantum Approximate Optimization Algorithm. The main difference between the two concerns the problem constraints. While the Quantum Approximate Optimization Algorithm incorporates them through penalty terms in the objective function, the Quantum Alternating Operator Ansatz incorporates them directly in the mixing operators and the initial state. The acronym "QAOA" is used for both approaches in their original papers and in further literature. Often it is not clearly distinguished between the two and Quantum Alternating Operator Ansatz is used as an umbrella term.
> We use QAOA as an acronym for the Quantum Alternating Operator Ansatz and explicitly distinguish the Quantum Approximate Optimization Algorithm when necessary.
>
> **More Detailed Explanation of Correlation Clustering**
>
> Due to the character limit, we kindly refer the reviewer to the answer given in the rebuttal to reviewer fb8D.
>
>
> **Translation of Correlation Clustering to SQAOA**
>
> Due to the character limit, we kindly refer the reviewer to the answer given in the rebuttal to reviewer fb8D.
>
> **Applications beyond Correlation Clustering**
>
> Due to the character limit, we kindly refer the reviewer to the answer given in the rebuttal to reviewer L5UV.
>
> **Modelling with Qubits instead of Qudits**
>
> The reviewer states that qudits are a perfect fit for clustering multiple labels (and thus for correlation clustering) and sees no motivation for our formulation of the problem in terms of qubits. We agree that qudits are a natural choice for correlation clustering. At the same time, our SQAOA formulation based on qubits outperforms the qudit-formulation in terms of approximation ratios and runtimes, while requiring less advanced and less specialized hardware. We consider this contribution relevant also because it is not obvious.
>
> **Figure 1**
>
> To clarify the terms "agreement" and "probability" in Figure 1, we
> suggest to replace the caption of this figure by the following one:
>
> "Depicted are two diagrams showing the probability of measuring basis states when applying the multi-level QAOA formulation of Weggemans et al. (2022) with $p = 1$ to a correlation clustering problem instance on a complete graph with 4 nodes. Shown in addition are the agreements of these basis states, i.e. the value of the objective function in (6). The probabilities of the diagram at the top are obtained directly from the QAOA results. The probabilities of the diagram at the bottom are obtained by nucleus sampling with a threshold of $t = 0.5$.
>
> **Hyperparameter $t$**
>
> The nucleus size $t$ used for nucleus sampling is indeed a hyperparameter whose choice could be further evaluated.
> There is a trade-off between low probability states (for small t) and high sampling noise (for large t).
> Due to sampling noise, we expect the optimal parameter to decrease with increasing number of shots.
> However, even our ad hoc choice of t, without a systematic study, results in the greatly improved approximation ratios that we show in Table 1 and Appendix C.
>
> **Convergence to an Optimal Solution**
>
> The reviewer states that, while our theoretical claim of reaching an optimal solution when the ansatz depth tends to infinity is legit, it is of no surprise.
> All in all, we agree with this statement, especially taking into account the newly introduced term separating phases of individual qubits.
> At the same time, the convergence guarantee we establish is necessary, we think, to put our approach on the same theoretical foundation as the Quantum Alternating Operator Ansatz.
>
> **Comparison with Wegemanns et al.**
>
> Due to the character limit, we kindly refer the reviewer to the answer given in the rebuttal for reviewer L5UV.

---

### Official Review · Reviewer_fb8D · 2025-03-19

**Overall Recommendation:** 3

**Summary:**

The paper explores the power of the Quantum Alternating Operator Ansatz algorithm. In particular, the focus is on the optimization problem called correlation clustering and provides a way to solve it by dividing the Quantum Alternating Operator Ansatz algorithm in subproblems. They show that by iterating to infinity the optimal solution is found and provide experiments in specific topologies that show good performance.

## Update after rebuttal

Thank you to the authors for the answers to my questions. I think with the changes proposed the manuscript will improve.

**Claims And Evidence:**

The claim is that Quantum Alternating Operator Ansatz algorithm can be adapted to find the optimal solution to the correlation clustering. This is shown analytically and empirically.

**Essential References Not Discussed:**

I did not find anything missing, but I would like to see a discussion on the difference between Quantum Approximate Optimization Algorithm and Quantum Alternating Operator Ansatz, and how one generalizes the other.

**Experimental Designs Or Analyses:**

I looked at the empirical results and they seem fine.

**Methods And Evaluation Criteria:**

The analysis seems to be correct. The empirical evaluation is restricted to some specific topologies, but they are general enough for a first study.

**Other Comments Or Suggestions:**

Say what is p in the abstract.

I feel that the correlation clustering problem statement was a bit underexplained (lines 165-178), and I did not quite understand how the original problem is transformed into the SQAOA, and how the max-cut problem is used to construct that definition, I think a bit more detail on the construction of equation 11, and the use of the gate of equation 9 could be useful.

**Other Strengths And Weaknesses:**

The proposed approach for the Quantum Alternating Operator Ansatz can possibly be used for other problems. However, for now the application is only for correlation clustering, making it limited.

**Questions For Authors:**

What is the difference between Quantum Alternating Operator Ansatz and Quantum Approximate Optimization Algorithm? Wy is the same acronym used for both?

How specific is your subproblem decomposition for the correlation clustering problem? Can it be applied to other problems? Which are good candidates? (You say "splitting a problem in sub-problems is a universal approach, and similar improvements might be possible on other problems suitable for QAOA.")

**Relation To Broader Scientific Literature:**

QAOA is usually the acronym for Quantum Approximate Optimization Algorithm. I think this has to be mentioned in the introduction.

**Theoretical Claims:**

I did not check the proofs in detail, but the results and claims are intuitively correct.

---

> ### Author Rebuttal · Authors · 2025-03-31
>
> We thank all reviewers for their constructive feedback.
> There seems to be consensus that our experimental and theoretical claims are correct and that the proposed approach surpasses previous QAOA methods for correlation clustering. Here, we gladly address remaining questions and concerns of Reviewer fb8D:
>
> **QAOA**
>
> The Quantum Alternating Operator Ansatz is a generalization of the Quantum Approximate Optimization Algorithm that was established after it. The acronym "QAOA" is used for both approaches in their original papers and in further literature.
>
> Differences between the two approaches concern the incorporation of constraints and the use of problem-dependent mixing operators and initial states: On the one hand, the Quantum Approximate Optimization Algorithm first transforms the given optimization problem into an equivalent one in which the constraints are replaced by penalty terms in the objective function. Then, fixed mixing and problem-dependent phase-separation operators are applied alternatingly to the initial state that is an equal
> superposition of all possible states. On the other hand, the Quantum Alternating Operator Ansatz works directly on the original problem by incorporating the constraints in problem-dependent mixing operators and by choosing an initial state that is a superposition of feasible states.
>
> **Applications beyond Correlation Clustering**
>
> Due to the character limit, we kindly refer the reviewer to the answer given in the rebuttal to reviewer L5UV.
>
> **$\mathbf{p}$ in the Abstract**
>
> We agree that our use of the symbol $p$ in the abstract needs clarification and suggest to replace "if $p \rightarrow \infty$" with "if the depth of the ansatz tends to infinity".
>
> **More Detailed Explanation of Correlation Clustering**
>
> To further illustrate the ILP formulation of the correlation clustering problem, we propose to insert the following paragraph after Line 178 and to also include a figure with a simple example.
>
> "In this formulation, the variable assignment $x_{u,1}=1$ and $x_{v,2}=1$ for nodes $u,v$ indicates that node $u$ is in Cluster $1$ and that node $v$ is in Cluster $2$. Thus, the nodes are in different clusters. A value of $1$ is contributed to the objective value if and only if $c_{uv}=-1$."
>
> **Translation of Correlation Clustering to SQAOA**
>
> To comment further on the construction of the phase-separation operator, we propose to insert the following paragraph after Line 138:
>
> "In order to apply QAOA to a specific problem, one needs to define and implement the operators and the initial state. The main challenge here lies in the construction of the initial state and mixing operator. Given a binary ILP formulation of the problem, as in (6), the phase-separation operator can be constructed easily by replacing a variable $x$ in the cost function $C(x)$ by the term $\frac{(1-Z)}{2}$ in the phase-separation Hamiltonian $H_C$. This is due to the fact that if $x = 0$ then $\frac{(1-Z)}{2} |x\rangle =
> 0 |x\rangle$, and if $x = 1$ then $\frac{(1-Z)}{2} |x\rangle = 1 |x\rangle$, and thus, (2) is fulfilled for the Hamiltonian constructed
> in this way. Implementing the corresponding unitary operator, then only requires the application of rotational Pauli-Z gates to individual qubits."
>
> To explain in more detail the specific operators used in our SQAOA formulation of correlation clustering, we suggest to insert the following paragraph after Line 248:
>
> "The transition operator $U_{T_i,x}$ uses Hadamard gates to construct an equal superposition of all feasible states of the sub-problem. The mixing operator $U_{M_i,x}$ enables transitions between the feasible states of a sub-problem by flipping qubits, i.e. by changing if the corresponding nodes remain in the current cluster, or are assigned to a new cluster that is further split-up in the next sub-problem. The phase-separation operator $U_{C_i,x}$ incorporates in the first sum of the exponent the cost function as described in Section 3 but drops constant terms. Additionally, the Hamiltonian given by the second sum of the exponent allows a cost-independent separation of phases based on individual vertices."
>
> **Sub-Problem Decomposition**
>
> The decomposition we define is specific to the correlation clustering problem. Other decompositions are conceivable and might yield further improvements. In general, the main challenge in applying the Quantum Alternating Operator Ansatz lies in the definition and implementation of mixing operators and initial states appropriate for the specific optimization problem. For applying SQAOA, in addition, the sub-problems need to be chosen carefully.
>
> Beyond correlation clustering, candidates for our sub-problem approach are ones in which elements are assigned one of multiple labels. E.g., one could consider the Maximum $k$-Colorable Subgraph Problem with sub-problems coloring so-far-unconsidered parts of the graph by a fixed number of colors smaller than $k$.

---

### Decision · Program_Chairs · 2025-05-01

**Decision:**

Accept (poster)

**Comment:**

The paper presents a new quantum optimization approach called the Sub-Problem Quantum Alternating Operator Ansatz (SQAOA) aimed at solving correlation clustering (CC) problems, which present a new kind of problem that can be solved by the general family of QAOA.  It is a solid contribution. However,  the presentation could use extra efforts: in particular, it would be helpful to include more explanation and motivation for the CC problem, a broader literature review and a more thorough discussion of the classical computational limitation.